



# Exploring the Drivers of Tropospheric Hydroxyl Radical Trends in the GFDL AM4.1 Atmospheric Chemistry-Climate Model

Glen Chua[1,2], Vaishali Naik[2], and Larry Horowitz[2]

[1]Princeton University, Program in Atmospheric and Oceanic Science
[2]NOAA Geophysical Fluid Dynamics Laboratory, Princeton, NJ, USA

**Correspondence:** Glen Chua (gchua@princeton.edu)

**Abstract.**

We explore the sensitivity of modelled tropospheric hydroxyl (OH) concentration trends to meteorology and near-term climate forcers (NTCFs), namely methane ($CH_4$); nitrogen oxides ($NO_x = NO_2 + NO$); carbon monoxide (CO); non-methane volatile organic compounds (NMVOCs); and ozone-depleting substances (ODS) using the Geophysical Fluid Dynamics Laboratory (GFDL)'s atmospheric chemistry-climate model, Atmospheric Model version 4.1 (AM4.1) driven by emissions inventories developed for the Sixth Coupled Model Intercomparison Project (CMIP6) and forced by observed sea surface temperatures and sea ice prepared in support of the CMIP6 Atmospheric Model Intercomparison Project (AMIP) simulations. We find that the modelled tropospheric airmass-weighted mean [OH] has increased by $\sim 5\%$ globally from 1980 to 2014. We find that $NO_x$ emissions and $CH_4$ concentrations dominate the modelled global trend, while CO emissions and meteorology were also important in driving regional trends. Modelled tropospheric $NO_2$ column trends are largely consistent with those retrieved from the Ozone Monitoring Instrument (OMI) satellite, but simulated CO column trends generally overestimate those retrieved from the Measurements of Pollution in The Troposphere (MOPITT) satellite, possibly reflecting biases in input anthropogenic emission inventories, especially over China and South Asia.

## 1 Introduction

The hydroxyl radical (OH), as the primary daytime oxidant in the troposphere (Levy, 1971), plays an important role in atmospheric chemistry. OH influences air quality and climate, as reaction with OH is a major sink of various trace species including tropospheric ozone precursors such as methane ($CH_4$), carbon monoxide (CO), nitrogen oxides ($NOx = NO + NO_2$), ozone-depleting substances (ODS) such as halocarbons, and non-methane volatile organic compounds (NMVOCs) (e.g. Holmes et al. (2013); Turner et al. (2019)). Precise knowledge of the OH budget, its variations and trends, and its response to various drivers, is needed to determine source and sink budgets for these important trace species, and is therefore crucial to our understanding of the various aforementioned effects on the Earth system (e.g. Lawrence et al., 2001; Naik et al., 2013; Murray et al., 2014; Zhao et al., 2019; Nicely et al., 2020; Patra et al., 2021). In particular, there are still gaps in our understanding of the drivers of the observed atmospheric methane concentration growth rate in recent years (Saunois et al., 2020; Nisbet et al., 2021), further highlighting the importance of better understanding [OH] trends and variability. Recent observational based studies suggest





**Table 1.** Important reactions describing tropospheric OH chemistry.

| Reaction number | Reaction |
|---|---|
| R1 | $O_3 + h\nu \rightarrow O(^1D) + O_2, \lambda < 310nm$ |
| R2 | $O(^1D) + H_2O \rightarrow 2OH$ |
| R3 | $HO_2 + NO \rightarrow NO_2 + OH$ |
| R4 | $O_3 + HO_2 \rightarrow 2O_2 + OH$ |
| R5 | $H_2O_2 + h\nu \rightarrow 2OH, \lambda < 550nm$ |
| R6 | $CO + OH(+O_2) \rightarrow HO_2 + CO_2$ |
| R7 | $RH + OH(+O_2) \rightarrow RO_2 + H_2O(+O_2)$ |
| R8 | $RO_2 + NO(+M) \rightarrow R'CHO + NO_2 + OH(+M)$ |
| R9 | $NO_2 + h\nu \rightarrow O(^3P) + NO, \lambda < 430nm$ |
| R10 | $O(^3P) + O_2(+M) \rightarrow O_3(+M)$ |
| R11 | $NO_2 + OH(+M) \rightarrow HNO_3(+M)$ |
| R12 | $HO_2 + HO_2 \rightarrow H_2O_2 + O_2$ |
| R13 | $RO_2 + HO_2 \rightarrow ROOH + O_2$ |

either a decline or stable OH concentrations over the past four decades (e.g. Rigby et al., 2017; Turner et al., 2017; Zhao et al., 2019) while global chemistry-climate models simulate increases over the same period (e.g Stevenson et al., 2020; Zhao et al., 2020). In this study, we employ the state-of-the-science GFDL chemistry-climate model (CCM), AM4.1, to systematically explore the drivers of changes in [OH] between 1980 and 2014 to shed light on its role in driving recent methane increases.

Changes in [OH] can be traced back to changes in the budget terms. The time tendency of [OH] is determined by the

balance between chemical production (P) and loss (L) terms, since the chemical processes for OH tend to occur at much faster timescales compared to other potential terms in the budget equation such as advection and transport (Lelieveld et al., 2016). The governing time tendency equation therefore is given by

$$\frac{d[OH]}{dt} = P - L \tag{1}$$

Table 1 summarises the tropospheric OH chemistry that is described here. Primary production of tropospheric OH occurs

via the photodissociation of tropospheric ozone ($O_3$) by ultraviolet (UV) radiation of wavelength less than 310 nm (R1) to produce excited singlet oxygen atoms ($O(^1D)$) (Brasseur and Solomon, 2005), which then react with water vapor (R2). OH can also be generated by secondary production mechanisms that recycle OH from hydroperoxy radicals ($HO_2$) (reactions R3-R5). In high-NOx regions, such as polluted urban environments, $HO_2$ can be recycled back to OH via reaction with NO without consuming $O_3$ (R3) and is the dominant production term. This $NO_x$-driven secondary prodution of OH, otherwise known as

the $NO_x$ recycling mechanism of OH, is similar in magnitude to the primary formation of OH on a global basis, being about $\sim 30\%$ each (Lelieveld et al., 2016). In unpolluted regions, other secondary production mechanisms, collectively called the $O_x$ recycling mechanism, are dominant. One involves the consumption of ozone in unpolluted regions (R4) (as opposed to





the production of ozone in polluted conditions), and the other involves the photolysis of $H_2O_2$ (R5). Oxidation of CO is the

largest OH loss reaction (R6) with important losses via oxidation of methane and non-methane volatile organic compounds

(NMVOCs) (R7). Organic peroxy radicals can also undergo an OH-recycling reaction with NO to form $NO_2$ (R8). The fate of

the resultant organic carbon product, $R'CHO$, can be to either further generate OH or $HO_2$ radicals if it undergoes photolysis,

or to undergo further oxidation by OH. The $NO_2$ produced via Reactions R3 and R8 can then be photolysed to form ozone

(R9-R10), which then leads to further primary production of OH via Reactions R1 and R2 (Hameed et al., 1979). In a strongly

polluted atmosphere, $NO_2$ can locally become a large $HO_x$ ($HO_x = OH + HO_2$) sink, causing net OH loss through the formation

of nitric acid ($HNO_3$) (R11) which can be washed out via wet deposition (Crutzen and Lawrence, 2000). Meanwhile, in clean,

non-polluted conditions, the reaction chains involving the $HO_2$ and $RO_2$ radicals can be terminated via loss reactions R12 and

R13. The self-reaction of $HO_2$ (R12) represents an OH sink, as the hydrogen peroxide ($H_2O_2$) product can be washed out via

wet deposition and is the dominant $HO_x$ sink since most of the troposphere experiences low NOx conditions (Jaeglé et al.,

2001).

Overall, the atmospheric composition directly impacts the OH budget, most notably via tropospheric ozone, humidity, $NO_x$,

CO, methane and NMVOCs, with the former three usually acting to increase [OH], and the latter three acting to decrease [OH],

but there are also meteorological factors which influence the budget via influencing the tropospheric chemistry of OH as well.

Temperature plays an important role in controlling rate reaction rates, tropospheric water vapour abundance and also natural

emissions of biogenic VOCs (Spivakovsky et al., 2000). Also, as many important reactions are photolysis reactions, such as

the primary production of OH via R1 which requires UV radiation of wavelengths ($\lambda < 330$nm), the overhead ozone column,

which controls the amount of UV radiation penetrating into the troposphere, aerosol direct and indirect effects, and cloud cover

play an important role as well (Levy, 1971). These point to the possible anthropogenic impacts on the OH budget which can

impact these various factors directly or indirectly.

Because OH is highly reactive and therefore has a short lifetime of $\sim 1$s, this makes it difficult to achieve global observational

coverage over time of directly-observed [OH]. As a result, various observational proxies have been used to indirectly estimate

the spatial distribution, global mean as well as the temporal variations and trends of [OH]. A widely-used proxy is methyl

chloroform ($CH_3CCl_3$, MCF) (e.g. Montzka et al., 2011; Rigby et al., 2017; Turner et al., 2017; Naus et al., 2019; Patra et al.,

2021), for which there is a relatively long temporal observational record, for example the $\sim 4$ decades' worth of data from the

Advanced Global Atmospheric Gases Experiment (AGAGE) and National Oceanic and Atmospheric Administration (NOAA)

networks. Using a multi-box model inversion method, Rigby et al. (2017) and Turner et al. (2017) found an increasing global

mean [OH] trend from the 1990s up to the mid-2000s but found a decreasing [OH] trend thereafter; however, in these studies,

it was highlighted that the inferred [OH] trends were only weakly constrained, and when Naus et al. (2019) corrected for biases

in the multi-box model inversion method, they found an overall increasing trend over the last 2 decades. To avoid some of these

biases such as those arising from the spatial averaging required in box model methods, 3D chemistry transport models (CTMs)

have also been used to infer [OH] from MCF observations such as in Patra et al. (2021) and Naus et al. (2021) who found

no significant trend in [OH]. Non-MCF methods have also been used to explore [OH] trends since 1980: for example, Nicely

et al. (2018) used observational constraints of various [OH] drivers to empirically reconstruct [OH], and found no significant





trend as well. Overall, global mean [OH] derived from most atmospheric inversions or empirical reconstructions seems to not have a significant trend over the 1980-2014 period. Models such as CTMs, CCMs and earth system models (ESMs) can also be

used to calculate [OH], and these models have shown an increase in global mean [OH] since 1980 to present-day, in contrast to the [OH] trends derived from observational constraints (Naik et al., 2013; Zhao et al., 2019; Nicely et al., 2020; Stevenson et al., 2020; Zhao et al., 2020). Most of these studies found that changes to Near-term Climate Forcers (NTCFs) played a key role in driving the modelled [OH] increase. In particular, Stevenson et al. (2020) found an $\sim 10\%$ increase with respect to the 1998-2007 mean from 1980 to 2014 from 3 Earth System Models (ESMs) participating in the Aerosols and Chemistry Model

Intercomparison Project (AerChemMIP) as part of the Sixth Coupled Model Intercomparison Project (CMIP6). They attribute this simulated increasing to changes in anthropogenic NTCFs, mainly increases in anthropogenic nitrogen oxides combined with declining CO emissions since 1990 with smaller contributions from changes in halocarbon and aerosol-related emissions. Naik et al. (2013) and Nicely et al. (2020) also additionally highlight the role of stratospheric ozone loss due to factors such as emissions of ozone-depleting substances (ODS) and increasing specific humidity in driving the increasing [OH] trend.

As discussed above, there are a plethora of emissions-related, chemical and physical drivers that affect [OH], and many of these are also driven by climate variability (Alexander and Mickley, 2015). As such, [OH] tends to exhibit interannual variability (IAV), and various modelling studies have explored the drivers of [OH] IAV. For example, large-scale climate variability through the El Niño Southern Oscillation (ENSO) has been shown to play an important role in driving [OH] IAV, through ENSO effects on variability in: temperature and humidity (Zhao et al., 2020); biomass burning emissions such as CO

(an OH sink) and $NO_x$ (tends to enhance OH) (Holmes et al., 2013; Zhao et al., 2020); $O_3$ and $j(O^1D)$ in the lower troposphere (Anderson et al., 2021); and lightning $NO_x$ emissions (Turner et al., 2018). In particular, the role of lightning $NO_x$ in driving [OH] IAV in the GFDL AM4.1 over the period 1980-2016 was also highlighted by He et al. (2021), and Murray et al. (2013) who found that lightning $NO_x$ emissions were the key factor driving IAV especially over the period 1998-2006 through its role in affecting both seconday OH production via the $NO_x$ recycling mechanism as well as primary production via its role as a

tropospheric ozone precursor. However, the drivers of OH variability also show large model diversity. For example, lightning $NO_x$ can be parameterised differently in different models (Zhao et al., 2020; Wild et al., 2020), and so its response to climate variability like ENSO can vary from model to model.

In summary, in the period 1980-2014, global CCMs seem to have converged on an overall increasing [OH] trend driven by complementary changes in emissions and meteorology (Szopa et al., 2021). Here we build on previous studies to explore

the contribution of individual component drivers to attribute trends and variability in [OH]. We apply the the GFDL-AM4.1 CCM to systematically explore the roles of meteorology and individual chemical drivers in changing [OH], with the goal of identifying the primary drivers of increasing [OH] trends over 1980-2014 simulated by global models. Additionally, we analyse the model simulations to shed light on the primary drivers of [OH] IAV.





## 2 Methods

### 2.1 GFDL AM4.1 Model Setup

We use the GFDL Atmospheric Model 4.1 (AM4.1), which is the atmosphere-only configuration of the GFDL Earth System Model ESM4.1. Further details of the AM4.1 setup are described by Horowitz et al. (2020), and a summary of the features relevant to [OH] are provided here. The AM4.1 has a spatial resolution of $\sim 100$km on a cubed-sphere grid, and resolves 49 vertical levels up to $\sim 80$km. It uses an updated chemical mechanism (Horowitz et al., 2020) with gas-phase and heterogeneous chemistry updates following Mao et al. (2013a, b). The key feature of this model configuration is that it has online oxidants, i.e. it includes chemical and climate feedbacks on oxidant concentrations. Photolysis rate constants are calculated interactively via the photolysis mechanism Fast-JX version 7.1 (Wild et al., 2000; Bian and Prather, 2002). All model simulations are forced with interannually varying sea surface temperatures and sea ice from Taylor et al. (2000), prepared in support of the CMIP6 Atmospheric Model Intercomparison Project (AMIP) simulations.

We used historical emissions datasets for ozone and aerosol precursors developed in support of phase 6 of the Coupled Model Intercomparison Project (CMIP6): the Community Emissions Database (CEDS) for anthropogenic emissions (v2017-05-18; Hoesly et al. (2018)) and BB4CMIP for biomass burning emissions (van Marle et al., 2017). In addition, natural sources of NMVOCs, $NO_x$ and CO were taken from Precursors of Ozone and their Effects in the Troposphere inventory (POET, Granier et al. (2005)) following Naik et al. (2013), except for isoprene and monoterpene emissions which are calculated online as described in Horowitz et al. (2020) and Rasmussen et al. (2012). Lightning $NO_x$Horowitz et al. (2020) emissions are calculated interactively as a function of subgrid-scale convection (Horowitz et al. (2020)). Well-mixed greenhouse gas concentrations are specified following Meinshausen et al. (2017). In particular, atmospheric concentrations of ODS, including CFC-11, CFC-12, CFC-113, and HCFC-22, and $CH_4$ concentrations are specified at the surface as a lower boundary condition, with concentrations beyond the surface subsequently determined by various chemical and dynamical processes. A summary of historical CO emissions, $CH_4$ concentrations and $NO_x$ emissions are shown in Fig. 1.

### 2.2 Model Runs

We conducted model integrations from 1980-2014, using an initialisation state from an GFDL AM4.1 historical run. In addition to a 'Base' run which includes the time-varying historical emissions of the various species as per Horowitz et al. (2020), we conducted 'all-but-one' runs where we investigate the effects of various emitted species which could affect OH concentrations, namely: $NO_x$; $CH_4$; CO; NMVOC; and ozone-depleting substances ('ODS'). These runs are configured so as to systematically fix the emissions of a particular species to 1980 values in order to isolate the effects of each individual species by comparing with the 'Base' run. Additionally, we include a run where all the above species are set to 1980 values ('Met'), which allows us to diagnose the effects of meteorology. Note that, as lightning $NO_x$ emissions as well as biogenic terpene and isoprene emissions are interactively-calculated, their impacts are included in the 'Met' run. The '$NO_x$' run only fixes non-lightning $NO_x$ emissions at 1980 levels, and the 'NMVOC' run only fixes the emissions of other NMVOCs than biogenic terpene and isoprene.





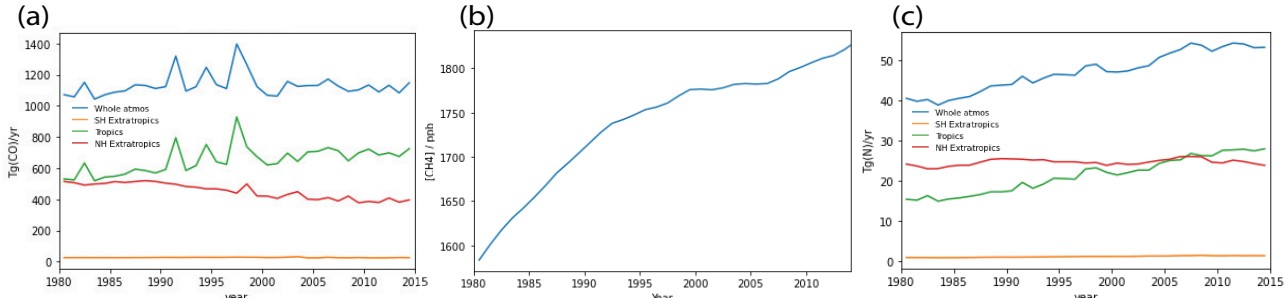

**Figure 1.** Historical time series of (a) regional CO emissions, (b) $CH_4$ surface concentrations, and (c) regional $NO_x$ emissions used for this study. The Southern Hemisphere (SH) extratropics is defined as $90^\circ S - 30^\circ S$, the Tropics as $30^\circ S - 30^\circ N$, and the Northern Hemisphere (NH) extratropics as $30^\circ N - 90^\circ N$. Globally, $CH_4$ concentrations have inceased by $\sim 16\%$. Non-lightning $NO_x$ emissions have also increased by $\sim 30\%$ globally over this period, with this increase driven by the Tropics. CO emissions see IAV but little trend globally over this period, but do see a decreasing trend in the NH extratropics offset by an increasing trend in the Tropics.

While the 'Base' and 'Met' run will be analyzed on their own, for the sensitivity runs involving each emission driver, $i = NO_x$, $CH_4$, CO, VOC, ODS, for each quantity analysed, e.g. tropospheric airmass-weighted OH concentrations, we calculate a derived quantity as per Eq. 2. $\text{Quantity}_i$ will be the quantity if the emission driver $i$ was set to 1980 values, but everything else (other drivers and meteorology) was as per the 'Base' run, so taking the difference $(\text{Quantity}_{Base} - \text{Quantity}_i)$ allows us to isolate the impact of driver i, with respect to the 1980 value from the 'Base' run $(\text{Quantity}_{Base,1980})$ and removes the impact of meteorology and other drivers. Adding the anomaly to the 1980 value from the 'Base' run then allows us to diagnose the impact that the emission driver $i$ would have on the quantity in isolation.

$$\text{Quantity}_{i,derived} = (\text{Quantity}_{Base} - \text{Quantity}_i) + \text{Quantity}_{Base,1980} \tag{2}$$

In the original run, $\text{Quantity}_i$ shows a negative deviation from $\text{Quantity}_{Base}$, and also shows some variability as a result of other factors like meteorology, the derived quantity, $Quantity_{i,derived}$, would show a positive deviation from $\text{Quantity}_{Base}$ instead, in addition to having the variability from other factors like meteorology removed.

### 2.3 Chemical Budget Term Analysis

To complement our analysis of the main potential drivers of [OH] from 1980 to 2014, we provide a bottom-up, mechanistic understanding of how the various drivers affect [OH], by looking at the chemical budget. As described in Eq. 1, changes in [OH] can be traced to changes in the OH chemical production and loss terms. We follow the methodology of Lelieveld et al. (2016) to analyse the chemical budget terms. We group the production terms into: primary production (Reactions R1-R2); secondary production, or recycling, via $NO_x$ (Reaction R3); secondary production via $O_3$ (Reaction R4); secondary production via $H_2O_2$ photolysis (Reaction R5); and other OH production reactions (e.g. from recyling from peroxy radicals via Reaction R13 or other photolysis reactions.) Meanwhile, the loss terms are grouped into: loss via CO from Reaction R6; loss via $CH_4$





from Reaction R7; loss via NMVOC from Reaction R7; loss via $NO_y$, which includes loss via Reaction R11 but also via other nitrogen-containing species like $HNO_3$, $NH_3$ and nitrogen-containing isoprene oxidation products; loss via $HO_y$, which includes loss via $H_2$, O, $O_3$, $H_2O_2$, $HO_2$ and also self-reaction; and other loss reactions, which include loss to sulphur- and halogen-containing species.

## 2.4 Evaluation of Modelled CO and $NO_x$

We identified CO, non-lightning $NO_x$ emissions and $CH_4$ as key drivers of [OH] from 1980 to 2014, of which $CH_4$ concentrations are prescribed in the model. so it is important to look at how well the modelled CO and $NO_x$ compare to observations. This would allow us to say to what extent our findings from our model study could be generalised to the real world.

### 2.4.1 Comparison of Modelled CO column with Measurements of Pollution in The Troposphere (MOPITT) Satellite Observations

We evaluate the modelled tropospheric CO column trends against those measured by the MOPITT instrument, following Horowitz et al. (2020). The MOPITT V8 Joint (NIR+TIR) retrievals (Deeter et al., 2019) during 2001-2014 are used, which are available from the NASA Earthdata archive (https://earthdata.nasa.gov). The modelled CO column is interpolated to the same grid as the monthly MOPITT observations, and the averaging kernel is applied to the modelled monthly mean CO profiles following documentation provided by Deeter (2003) in order to compare between modelled and observed MOPITT CO columns.

Horowitz et al. (2020) previously calculated the seasonal climatological mean CO column in the GFDL AM4.1 and found a persistent model CO column low bias in the NH and high bias in the SH compared to MOPITT observations across seasons. Horowitz et al. (2020) also compared modelled surface CO concentration with measurements from a globally distributed network of air sampling sites maintained by the Global Monitoring Division (GMD) of the Earth System Research Laboratory at the National Oceanic and Atmospheric Administration (NOAA) (Pétron et al., 2019; data available at ftp://aftp.cmdl.noaa.gov/data/trace_gases/co/flask/) and the NH low bias/SH high bias was also seen in remote site comparisons. In this study, we complement the analysis by comparing the annual mean CO column trends.

### 2.4.2 Comparison of Modelled Tropospheric $NO_2$ column with Ozone Monitoring Instrument (OMI) Satellite Observations

We also identified $NO_x$ trends as driving the modelled increase in [OH], so it would be useful to find out if the $NO_x$ trends modelled matched well with observations. Compared to CO which has a lifetime of about a month, $NO_x$ has a relatively shorter tropospheric lifetime of $\sim$ day (Jacob, 2000), and so $NO_x$ burden is much more concentrated near emission areas. Therefore, an evaluation of tropospheric $NO_x$ observations will more readily give information about emissions and how they change with time. We analyse the $NO_2$ column trends from 2005 to 2014 to coincide with when OMI observations started, noting that the shorter time period may limit the trend analysis.



The OMI, which is onboard the Aura satellite, is a polar-orbiting, nadir-viewing, UV-visible spectrometer with a swath width of $2600 km$ and a nadir pixel size of $13 \times 24 km^2$. It observes backscattered solar radiation in the range of 270-500 nm with an average spectral resolution of 0.5 nm. It has a continuous data record since 1st Oct 2004, with global daily coverage for the first

3 years of operation, but since 25th June 2007, anomalous radiances have been observed in several of the pixel rows. These have been classified as the 'row anomaly' (http://projects.knmi.nl/omi/research/product/rowanomaly-background.php). Filtering for this row-anomaly problem could sometimes result in up to 50% field-of-view rejection rate, causing OMI to complete global coverage in 2 days instead of 1. More information about OMI v3 $NO_2$ product which will be used in this study can be found in Krotkov et al. (2017).

Although gridded data products are publicly available, they do not come with information necessary to calculate the averaging kernel, which, as discussed earlier, is necessary for a better comparison between retrievals and modelled quantities. In particular, for UV-visible measurements such as OMI, the tropospheric retrievals of $NO_2$ can be heavily influenced by aspects such as clouds, surface albedo, the presence of a stratospheric background and aerosols, as well the assumed a priori vertical profile (Eskes and Boersma, 2003), resulting in potentially large errors. The Differential Optical Absorption Spectroscopy

(DOAS) technique used is sensitive to the a priori vertical profile, and using the averaging kernels allows for the model-to-satellite comparisons to not be affected by systematic biases introduced by the a priori assumptions. This work will not use averaging kernels, and thus the comparisons between the model and OMI observations are only an approximation. We furthermore also note that the tropopause level used for processing the model data, which we determined using the WMO definition, is also different from that used in the OMI data. These factors add to the qualitative nature of the comparisons between the

model and OMI observations done in this study. However, despite these approximations, the purpose of this analysis is to show consistency between input $NO_x$ emission trends, modelled $NO_x$ trends, and observed trends.

## 3 Results

### 3.1 Tropospheric [OH] Trends from 1980 to 2014

As seen in Fig. 2(a), tropospheric airmass-weighted [OH] has increased by $\sim 5\%$ from 1980 to 2014 in the Base simulation.

The 1980-2014 period fits a linear trend of $0.033 \pm 0.06$ molec cm$^{-3}$yr$^{-1}$ (95% Confidence Interval (CI)) as seen from Fig. 2(d) and exhibits some IAV throughout the period. The simulated increase is well within the range found in Zhao et al. (2019) and on the upper bound of the range found in Naik et al. (2013). In terms of the anomaly with respect to 1998-2007 mean modelled $\sim 10\%$ agrees well with that simulated in ESMs participating in CMIP6 using the same emission drivers (Stevenson et al., 2020). In detail, we see that the 1980-2010 period is dominated by an increasing trend. From 2010 to 2014, we see a

slight decrease; however, as shown by He et al. (2020), looking further into 2015-2017, [OH] increases again, suggesting that the overall increasing trend is robust. The increase, especially in the period 2000-2010, is in contrast to some observational studies like Rigby et al. (2017) and Turner et al. (2017) who found a decrease instead. The increase over the 35-year period 1980 to 2014 is also in contrast to the period from 1870 to 1980, where [OH] did not exhibit a trend (Stevenson et al., 2020; Szopa et al., 2021). As seen in Fig. 2(b), the increase in [OH] occurs throughout the depth of the troposphere, but with the





largest increase seen from the surface to the lower troposphere. This could suggest that the increasing [OH] trend is driven by mainly surface drivers rather than, for example, lightning $NO_x$ emissions.

Next, we look at the sensitivty of simulated [OH] to various chemical drivers of [OH]. The simulated global tropospheric airmass-weighted [OH] are plotted in Fig. 2(c), with the individual model runs analysed as per Section 2.2. As seen in Fig. 2(d), the increasing non-lightning $NO_x$ emissions caused the largest positive [OH] trend of $0.041 \pm 0.004$ molec cm$^{-3}$yr$^{-1}$ (95%

CI), while there is a small positive trend arising from decreasing ODS concentrations of $0.005 \pm 0.0015$ molec cm$^{-3}$yr$^{-1}$ (95% CI). This suggests that the increasing non-lightning $NO_x$ emissions of $\sim 30\%$ ( Fig. 1(c)) have been the largest driving force behind the overall [OH] increase. However, the 'NO$_x$' run overestimates the positive trend, and fails to capture some of the modelled features in the 'Base' run, such as the dip in 1992 suggesting the important contributions of other factors which dampen the effect of increasing non-lightning $NO_x$ as we discuss below.

In terms of factors that contribute negatively to the [OH] trend, increasing $CH_4$ concentrations caused the largest negative trend of $-0.012 \pm 0.02$ molec cm$^{-3}$yr$^{-1}$ (95% CI), and there is also a small negative trend simulated in the 'NMVOC' run of $-0.003 \pm 0.02$ molec cm$^{-3}$yr$^{-1}$ (95% CI). The 'CH$_4$' run simulates a roughly 4% decrease from 1980 to 2014 as seen in Fig. 2(c), consistent with the increasing $CH_4$ concentrations seen over the period. We see that [OH] decreases from 1980 to 2000 before stabilizing up to 2007 after which it resumes its decrease. This follows the $CH_4$ trend, plotted in Fig. 1(b), seen over

this period, where $CH_4$ has increased from 1980 to 2000 before stabilizing up to 2007 after which it resumed its increase, such that the $CH_4$ burden has increased by about 16% by 2014 compared to 1980 values. Over the 1980-2014 period, CO emissions do not contribute to the global average OH trend but induce large IAV (see section 3.3) as evident in the CO simulation. This lack of OH trend is attributed to the lack of trend in CO emissions over the this 35 year period as seen in 1(a).

With $NO_x$ and $CH_4$ identified as the main factors affecting the global [OH] trend, we conducted an additional sensi-

tivity run accounting for their combined effects ('CH$_4$ + NO$_x$'). As seen in Fig. 2(d), the resultant OH trend of $0.032 \pm 0.05$ molec cm$^{-3}$yr$^{-1}$ (95% CI) matches the 'Base' run well, suggesting that the combined effects of $CH_4$ and $NO_x$ drive the overall modelled [OH] trend.

Other factors, such as meteorology, that have been known to drive [OH] do not show up strongly on the global tropospheric mean analysis. This result is consistent with He et al. (2021), who used AM4.1 driven by the same emissions as this study but

varied the meteorology field (model-calculated, NCEP and MERRA meteorology), and found that meteorology could affect the magnitudes of mean [OH], but not the trend. Nonetheless, these other factors could have important regional contributions, which we explore in the next section.

## 3.2 Regional Tropospheric [OH] Trends from 1980 to 2014

Next, we analyze the spatial patterns of the [OH] trends in order to get a more nuanced view of how [OH] is changing. In

the 'Base' run, the tropospheric airmass-weighted column mean [OH] increases over most areas, with the largest increases over much of Tropical Asia as well as China (Fig. 3). On the other hand, there are also areas such as over USA, some parts of Western Europe and Northern Russia, where there is a small decreasing trend, as well as Central Africa which sees a pronounced decreasing trend.





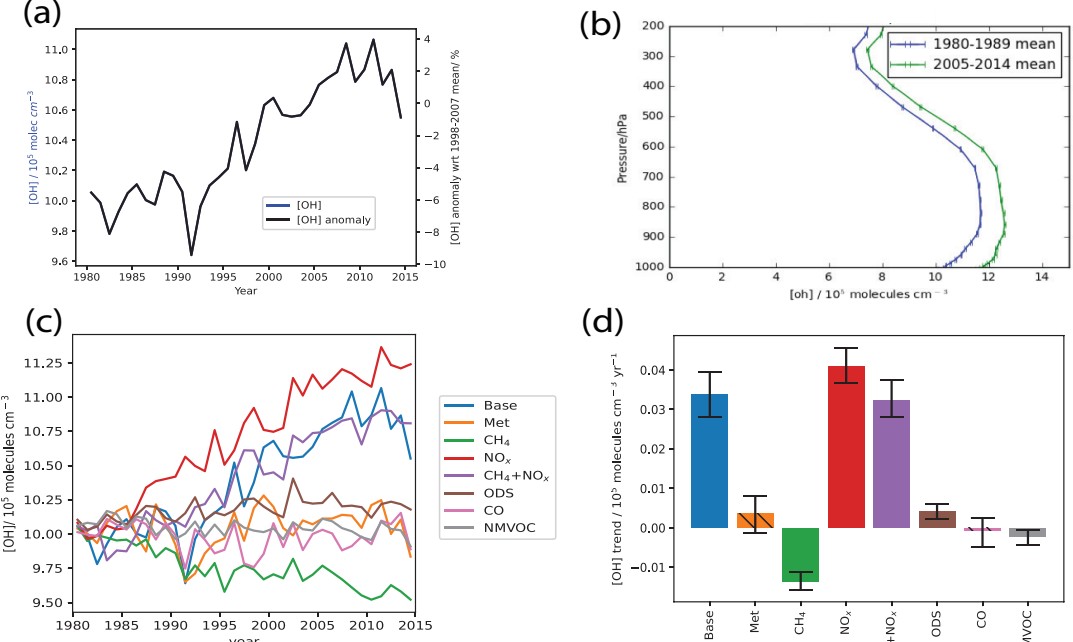

**Figure 2.** Tropospheric airmass-weighted [OH] (a) time series for the 'Base' run as well as the anomaly with respect to the 1998-2007 mean from 1980 to 2014; (b) 10-year area-weighted mean [OH] at various altitudes for 1980-1989 and 2005-2014 periods; (c) Tropospheric [OH] time series and (d) trends from the Base simulation and from the sensitivity simulations with all ('Met') and individual short-lived emissions held constant at 1980 levels. For (b), error bars represent $\pm 1$ standard deviation about the mean at each pressure level. For (d), trends are calculated using the Theil-Sen method, and error bars show the 95% confidence interval. Bars are hashed when no significant trend is detected at the 95% level using the Mann-Kendall test. From (a), tropospheric airmass-weighted [OH] has increased by $\sim 5\%$ from 1980 to 2014 in the Base simulation. As seen in (b), the increase in [OH] occurs throughout the depth of the troposphere, but with the largest increase seen from the surface to the lower troposphere, and this could suggest that the increasing [OH] trend is driven by mainly surface drivers. From (c) and (d), we see that globally the [OH] increasing trend is driven by the combined effects of $CH_4$ and $NO_x$.

We next see that the [OH] changes in the '$NO_x$' run are positively correlated with the changes in non-lightning $NO_x$
emissions (Fig. 4(a)), and itself matches the 'Base' run changes very well. This reinforces the earlier result that non-lightning
$NO_x$ emissions have been the main driver behind the modelled [OH] trend. However, non-lightning $NO_x$ emissions alone seem
to overpredict the decreasing trends over Western Europe and Northern Russia and increasing trends in the other regions. The
addition of the $CH_4$ trend does not change the spatial pattern much, since [$CH_4$] increases uniformly across the surface, but
it helps to bring the positive trends in line with 'Base'. However, this then leads to a larger overprediction of the decreasing
trends over Western Europe and Northern Russia.

This, in turn, is largely rectified by including the effects of CO emissions. We see that the 'CO' run produces changes in
[OH] that are negatively correlated with the changes in CO emissions (Fig 4(b)), in particular an increasing trend in [OH] over





USA, Europe and Russia associated with declining CO emissions. The addition of these effects therefore helps to dampen the effects of declining $NO_x$ emissions and increasing $CH_4$ concentrations in those regions. The dampening effect of CO on $NO_x$

effects in these particular regions is due to the large spatial correlation between the $NO_x$ and CO emission trends as seen in Fig. 4(a) and (b).

  The large negative trend over Central Africa results not only from contributions from increasing $CH_4$ concentrations and increasing NMVOC emissions (Fig. 4(c)), which tend to reduce [OH] by increasing chemical loss, but also from meteorology-related factors.

In terms of meteorology-related factors, trends in lightning $NO_x$ have been suggested to contribute to [OH] trends (e.g. He et al., 2020; Fiore et al., 2006). From Fig. 4(d), Our meteorology-driven simulation did not show significant lightning $NO_x$ trends except for a significant negative trend over Central Africa, which could further help explain the locally negative [OH] trend. Looking at other factors that could affect [OH], we see from Fig. 4(e) that isoprene emissions, which are interactively calculated in our model, have increased in specific regions in the meteorology-driven run, such as over the Amazon, Eastern

USA, Central Africa, and parts of Asia such as Eastern China. Indeed, we see that these regions are associated with negative [OH] trends in the meteorology-driven run, with Central Africa seeing a particularly large effect. We also see, from Fig. 4(f) that, in the meteorology-driven run, the water vapor burden has increased sigificantly in most regions, which tends to locally increase OH chemical production. These competing local effects on [OH] from meteorology-driven factors accounts for the absence of a globally-averaged [OH] trend contribution from meteorology.

Overall, this analysis reinforces the findings from the global average analysis of the key role of non-lightning $NO_x$ emissions in driving the overall [OH] trends, modulated by the changes in $CH_4$ concentrations. However, the regional trends highlight the additional importance of CO emissions, especially in regions in the extratropical NH, as well as the importance of increasing NMVOC emissions and meteorology-driven decreases in lightning $NO_x$ as well as increases in isoprene emissions in explaining the negative [OH] trend over Central Africa. Our findings from the spatial column tropospheric analysis also hold when we

looked at particular pressure levels in the surface as well as the middle and upper troposphere (Fig. A1).

### 3.3 Interannual Variability

In the above analyses, we found that, in general, $CH_4$ and $NO_x$ effects can largely explain the long-term [OH] trend over 1980-2014. However, the combined effects of $CH_4$ and $NO_x$ alone do not account for the short-term variability in [OH] seen over this period. In particular, there are some features such as the dip in 1992 as well as the dips and spikes seen between 1995

to 2000 that the '$CH_4$+$NO_x$' run misses. As seen in Fig. 5, in the global average there is a year-on-year change of up to about $0.4 \, \mathrm{molec \, cm^{-3} yr^{-1}}$, with 13 out the 34 years after 1980, or about one-third of the years, showing a negative year-on-year change.

  We see from Fig. 2(c) that the 'Base' and meteorology-driven runs are positively correlated, with a Pearson correlation coefficient of $r = 0.82$. As summarized in Section 1, meteorology can affect [OH] in various ways. We further see also that the

'CO' run (pink line) also exhibits variability features that are also seen in the 'Base' run, notably the dip in 1992, and also the

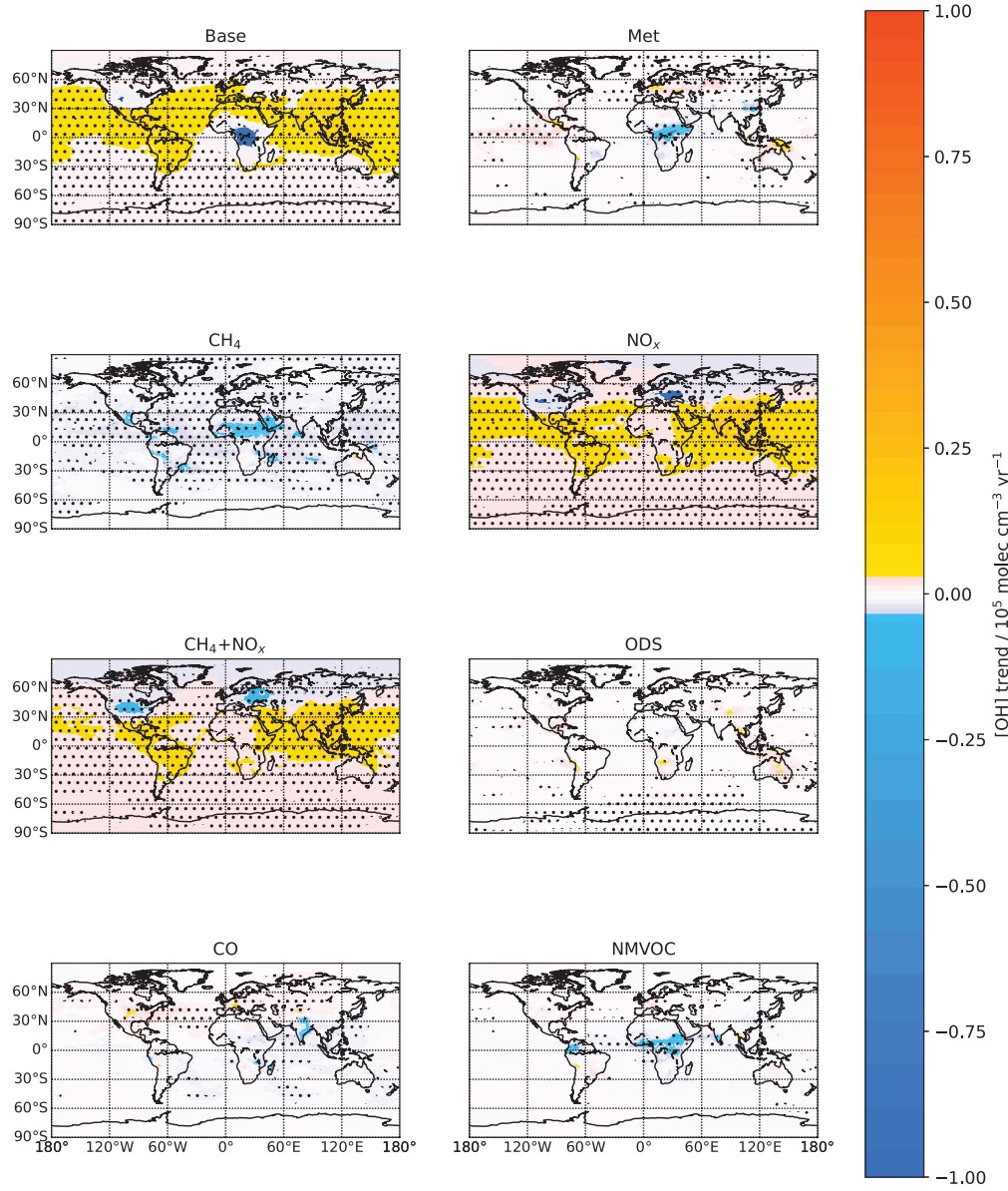

**Figure 3.** Trends in tropospheric airmass-weighted OH concentrations by grid box from 1980 to 2014 for the different model runs. Trends are calculated using the Theil-Sen method. Stipples show areas where a significant trend is detected at the 95% level using the Mann-Kendall test. In the 'Base' run, the tropospheric airmass-weighted column mean [OH] increases over most areas, with the largest increases over much of Tropical Asia as well as China. These increases are also largely driven by the combined effects of non-lightning $NO_x$ emissions and $CH_4$ concentrations. On the other hand, there are also areas such as over USA, some parts of Western Europe and Northern Russia, where there is a small decreasing trend, as well as Central Africa which sees a pronounced decreasing trend.



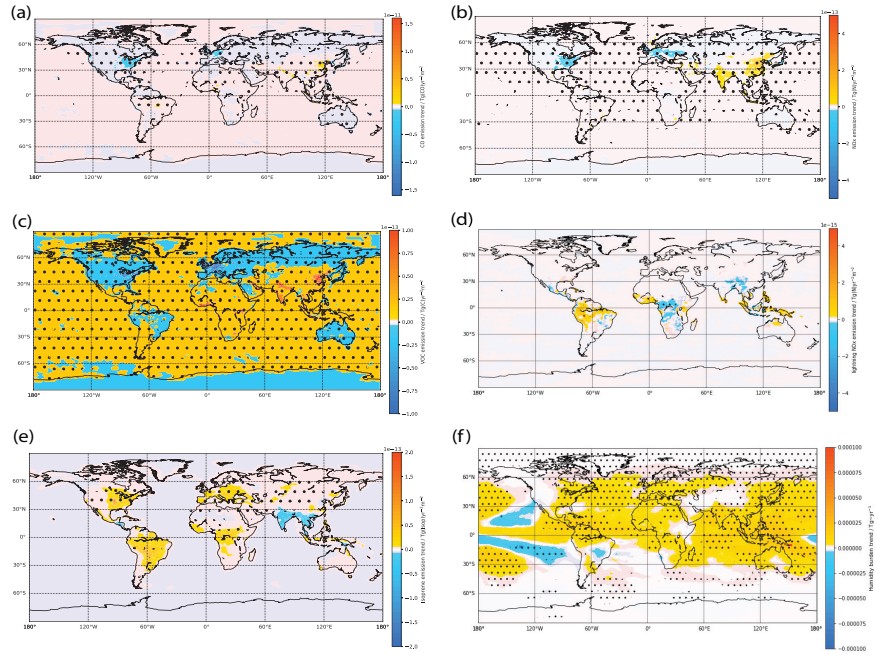

**Figure 4.** Spatial trends of (a) non-lightning $NO_x$, (b) CO, (c) NMVOC, (d) lightning $NO_x$ (e) biogenic isoprene emissions, and (f) humidity from 1980-2014. Trends and stippling are as per Fig. 3, except for (a) and (b) and (d), where we additionally remove stippling when the trend is below $10^{-16} Tg/yr^{-2} m^{-2}$. [OH] changes in the 'NO$_x$' run are positively correlated with the changes in non-lightning $NO_x$ emissions (seen in (a)) and itself matches the 'Base' run changes very well. Meanwhile, the 'CO' run produces changes in [OH] that are negatively correlated with the changes in CO emissions (seen in (b)). The large negative trend over Central Africa results not only from contributions from increasing $CH_4$ concentrations and increasing NMVOC emissions (seen in (c)), which tend to reduce [OH] by increasing chemical loss, but also from meteorology-related factors, such as lightning $NO_x$ as seen in (d), isoprene emissions as seen in (e). From (f), we see the water vapor burden has increased sigificantly in most regions, which tends to locally increase OH chemical production. These competing local effects on [OH] from meteorology-driven factors accounts for the absence of a globally-averaged [OH] trend contribution from meteorology.

dips and spikes seen between 1995 to 2000. This could be related to variability in biomass burning, as pointed out by Holmes et al. (2013).

## 3.4 OH production and loss terms

### 3.4.1 Chemical Budget Terms in Base Run

We first look at how the proportions of each of the production and loss terms evolves with time from 1980 to 2014 in the 'Base' run, as shown in Fig. 6(a) and (b). Firstly, we see that the relative proportions stay roughly constant throughout the time period, suggesting that all of the reaction terms have increased in tandem with the total. In terms of the biggest changes, we see that,





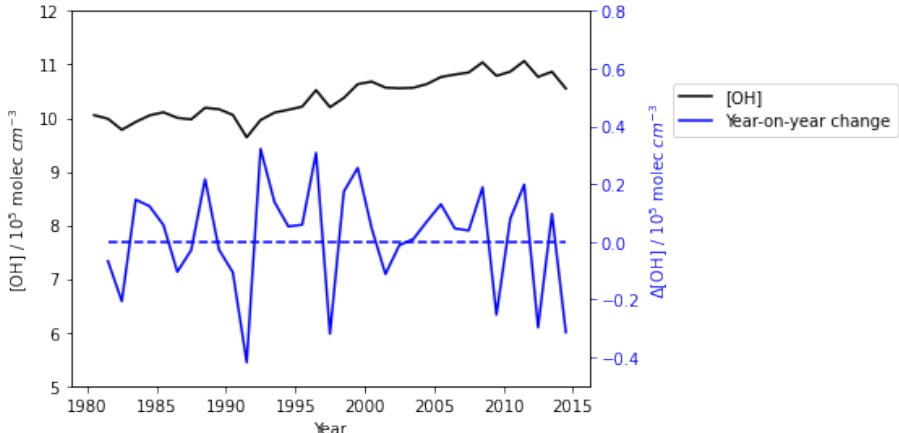

**Figure 5.** Base OH Concentrations plotted on the lefthand axis (black), and Year-on-Year changes plotted on the righthand axis (blue). In the global average there is a year-on-year change of up to about $0.4$ molec cm$^{-3}$yr$^{-1}$, with 13 out the 34 years after 1980, or about one-third of the years, showing a negative year-on-year change. The global IAV is driven by IAV in meteorology-related factors and CO emissions.

for OH production terms, the primary production term has increased in proportion by the largest amount (+0.5%), followed by small increases in the NO$_x$ and O$_3$ recycling reactions (0.1% each) at the expense of the other two terms. Meanwhile for the

OH loss terms, the CO loss reaction sees quite large variability, especially from 1990 to 2000, and decreases by 1.1% overall. The CH$_4$ loss reaction increases by 0.9% overall. Comparing the production values with Table 1 of Lelieveld et al. (2016), the percentages are roughly consistent. However, our model seems to have a larger proportion of primary production (42%) compared to Lelieveld et al. (2016) (33%). Looking at the loss reactions, we have a lower proportion of NMVOC loss (20% compared to 29%), and also a higher loss to HO$_y$ (25% compared to 18%).

We next look at the chemical budget terms in the 'Base' model run. From Fig. 6(c), we see that global tropospheric airmass-weighted OH production and loss match closely with one another, consistent with the pseudo steady-state assumption. Both have increased by about 14% by 2014 compared to 1980, showing a clear trend. We also see some IAV throughout the period, with year-on-year changes of up to 2%. Relating these results to the earlier findings, where we saw an increasing [OH] trend, the increasing trend in both production and loss (as they balance each other in steady-state) should therefore be driven by an

increasing trend in production, and that in turn should be associated with the net NO$_x$ and CH$_4$ effects. Meanwhile, the IAV observed in the OH concentrations should also be associated with the IAV seen in the production and loss, and this in turn should be affected by meteorological factors, factors that are driven by meteorology like lightning NO$_x$ and biogenic VOC emissions, and CO emissions. We delve into these hypotheses further in the following subsections when we do a sensitivity analysis for the budget terms.

Lastly, Fig. 6(d) shows the spatial plot of OH production trends from 1980 to 2014 for the 'Base' run. We see that it matches the spatial [OH] trend plot for the 'Base' run in Fig. 3, suggesting again the role of chemical production (and loss) in driving [OH] trends.





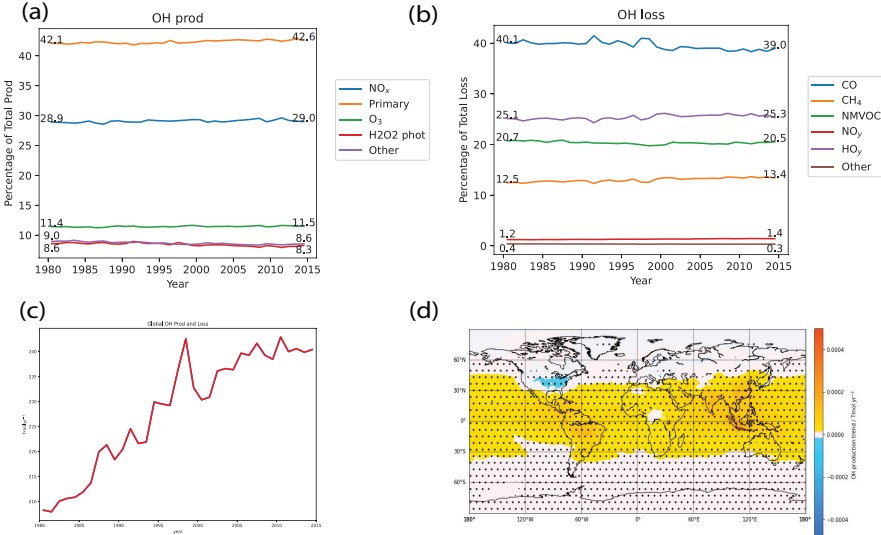

**Figure 6.** Tropospheric Airmass-weighted OH chemical production and loss terms in the 'Base' run. (a) Evolution of the proportion of each production reaction with respect to the total, and (b) Same but for loss reactions instead. (c) Time evolution of global total production and loss terms (curves overlap because OH is in pseudo steady-state.) (d) Spatial airmass-weighted tropospheric OH chemical production trend over the 1980-2014 period, with stippling and trends as per Fig. 3. From (a) and (b), we see that the relative proportions stay roughly constant throughout the time period, suggesting that all of the reaction terms have increased in tandem with the total, and the values are roughly consistent with Table 1 of Lelieveld et al. (2016). From (c), both global tropospheric airmass-weighted OH production and loss have increased by about 14% by 2014 compared to 1980, showing a clear trend. We also see some IAV throughout the period, with year-on-year changes of up to 2%. Relating these results to the earlier findings, where we saw an increasing [OH] trend, the increasing trend in both production and loss (as they balance each other in steady-state) should therefore be driven by an increasing trend in production, and that in turn should be associated with the net $NO_x$ and $CH_4$ effects. Meanwhile, the IAV observed in the OH concentrations should also be associated with the IAV seen in the production and loss, and this in turn should be affected by meteorological factors, factors that are driven by meteorology like lightning $NO_x$ and biogenic VOC emissions, and CO emissions. From (d), we see that spatial chemical production trends matches the spatial [OH] trend plot for the 'Base' run in Fig. 3, suggesting again the role of chemical production (and loss) in driving [OH] trends.

### 3.4.2 Chemical Budget Term Sensitivity Analysis

We next look at how these budget terms are affected by the input emissions by doing a sensitivity analysis using our model runs, by analyzing the sensitivity simulations involving each emissions driver as well as the meteorology-driven simulation. We focus our attention on the major terms represented in the budget. For production terms, we look at the primary production as well as secondary production via $NO_x$ and $O_3$, together accounting for $\sim 84\%$ of total production. For loss terms, we focus on loss via CO, $HO_y$ and $CH_4$, accounting for $\sim 78\%$ of total loss. Unfortunately, we did not have enough model diagnostics to study loss to NMVOCs, and this is left as potential future work. From the earlier analyses, the NMVOC-related budget term





is unlikely to play a large role in affecting the global [OH] trend, even though it may have a regional role. Also, while analysing the changes in these budget terms, we note that, due to the fact that [OH] is in pseudo steady-state, we will always have total production and loss approximately balancing at all times. This means that a change in production can precede a change in loss, or vice versa. To disentangle the effects, we have to rely on our physical understanding of the underlying chemistry, and can take cues from how [OH] itself is changing.

Since our analysis in section 3.1 showed the dominant role of $NO_x$ in driving [OH] increses over the 1980-2014 period, we focus here on $NO_x$. From Fig. 7 (c), we see from the red line that increasing $NO_x$ emissions have led to an increase in OH reycling from $NO_x$. This is to be expected, as the increasing $NO_x$ emissions as seen in Fig. 1(c) drive an increase in the tropospheric $NO_x$ burden as seen in Fig. A2(a). This therefore increases the $NO_x$ reaction recycling rate from Reaction R3, acting to increase the partitioning of $HO_x$ into OH. $NO_x$ emissions have also caused an increase in the other major production

terms. We can understand this via the impact of $NO_x$ emissions on tropospheric $O_3$. As seen in Fig. A2(b), increases in $NO_x$ emissions have driven the increasing trend seen in tropospheric $O_3$ burden, with the increase of $\sim 30\%$ of non-lightning $NO_x$ emissions leading to a $\sim 10\%$ increase in $O_3$. This suggests that the atmosphere as a whole is $NO_x$-limited with respect to ozone production, with ozone production occuring via Reactions R8 - R10. This is consistent with other studies e.g. Lawrence et al. (2003) who found that the lofting of surface $NO_x$ drove significant increases in $O_3$ production over the tropospheric

column. The increasing $O_3$ concentrations lead to an increase in primary production via Reactions R1 - R2. The increasing $O_3$ concentrations also lead to enhanced OH recycling via $O_3$ through Reaction R4, which further partitions $HO_x$ into OH. Murray et al. (2013) previously found that lightning $NO_x$ influenced both primary and secondary production, and we show here that non-lightning surface $NO_x$ emissions also have the same effect in our model. The increased OH production as a result of $NO_x$ emissions then leads to an increase in [OH]. As loss fluxes are proportional to [OH], this in turn then leads to increased OH

losses (as seen in Fig. 7 (b), (d) and (f)), eventually keeping OH in pseudo steady-state.

Next, we look at the impacts of $CH_4$, which we found to suppress the increasing [OH] trend. We first see the primary effect of increased $CH_4$ in depleting OH in Fig. 7(f). Furthermore, oxidation of $CH_4$ via Reaction R7 eventually leads to the production of CO as seen in Fig A2(c), which then leads to a small increase in OH loss via CO in Fig.7(b). However, as $CH_4$ is a tropospheric ozone precursor, such as via Reaction R8 - R10, the increase in $CH_4$ also leads to increased tropospheric $O_3$ as

seen in Fig.A2(b), thereby slightly enhancing primary production and OH recycling via $O_3$ as well. Hence, this could explain why the overall net negative effect of $CH_4$ on [OH], which includes a mixture of enhanced losses and production, is smaller. Furthermore, we see in Fig. 7(f) that the $NO_x$ run also contributes roughly equally to the OH loss flux to $CH_4$, and this further highlights the importance of changes in OH production associated with $NO_x$.

Lastly we look at the impacts of CO, which we found to have a regional effect on the [OH] trend. As identified earlier in Fig.

3, the main region where CO emissions have affected the [OH] trend is in some regions in the extratropical NH, such as Eastern USA and Western Europe, where a decrease in CO emissions led to an increase in [OH], and in South Asia and East China, where an increase in CO emissions led to a decrease in [OH]. As seen for the 'CO' run in Fig. 8, regions of decreasing CO emissions see decreasing OH loss via reaction with CO and hence an increase in [OH], and vice versa. However, comparing



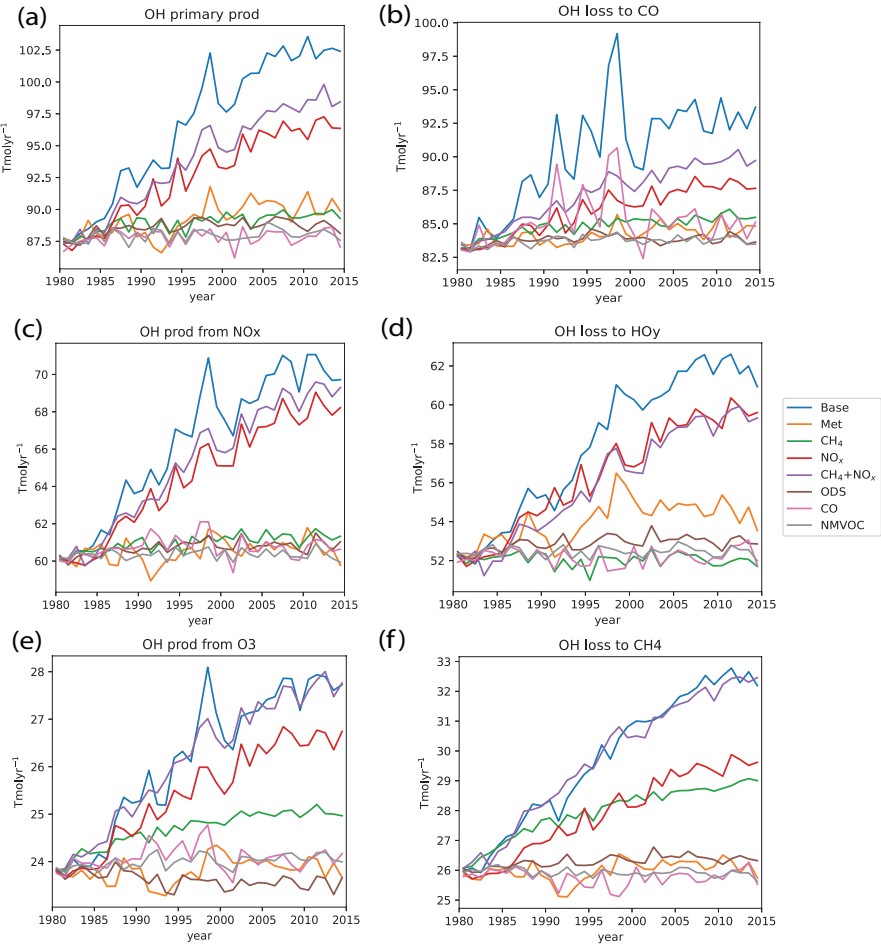

**Figure 7.** OH chemical production and loss budget terms for the different model runs. (a), (c) and (e) show the production terms, accounting for $\sim 84\%$ of total OH production. (b), (d) and (f) show the loss terms, and account for $\sim 78\%$ of total OH loss. Globally, increasing $NO_x$ emissions have led to increasing primary and secondary OH production, while increasing $CH_4$ concentrations have led to increased OH loss via reaction with $CH_4$ that is offset by increased secondary OH production due to the increase in tropospheric $O_3$.

the 'Base' and '$NO_x$' runs, we also see that the base OH loss flux to CO is also driven by the [OH] changes associated with

the $NO_x$ run, and this again highlights the importance of changes in OH production associated with $NO_x$.

Additionally, we find that meteorology plays an important role for IAV in OH primary production, while CO emissions are more important for IAV in the OH loss flux to CO. The former is evident from Fig. 9 which shows that OH primary production is strongly correlated with changes in specific humidity (q) (r = 0.90) which itself is strongly correlated with temperature (r = 0.96). For the latter, as seen in Fig. 10(a), CO emissions are positively correlated with OH loss to CO in the 'Base' run

($r = 0.69$). This thereafter drives overall IAV in total loss and production. As seen in Fig. 10(b), where we plot the year-on-year changes in [OH] on the lefthand axis (black), as well as the year-on-year changes in total OH production and loss and that of





**Figure 8.** Tropospheric airmass-weighted OH loss flux to reaction with CO spatial trends from 1980 to 2014 for the different model runs. Trends and stippling are as per Fig. 3. Regions of decreasing CO emissions see decreasing OH loss via reaction with CO and hence an increase in [OH] (as seen in the 'CO' run in Fig. 3), and vice versa.



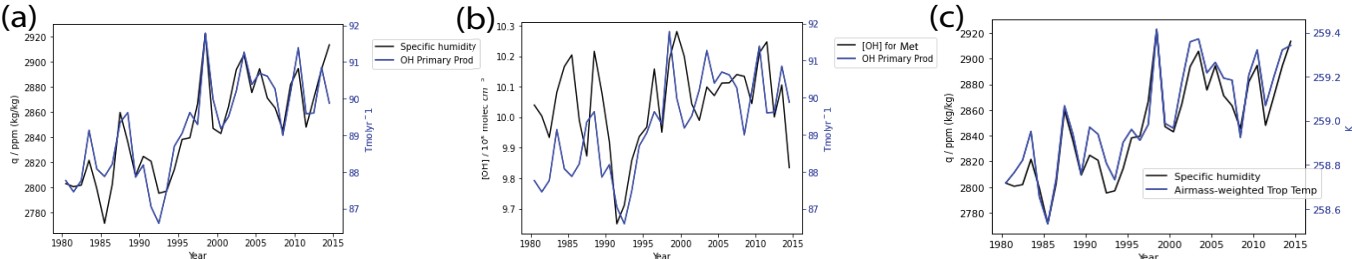

**Figure 9.** OH primary production and [OH] in the meteorology-driven ('Met') run. (a) shows the comparison between specific humidity (lefthand axis, black) and OH primary production (righthand axis, blue) in the meteorology-driven run; (b) shows the [OH] (lefthand axis, black) compared with OH primary production (righthand axis, blue); and (c) shows the comparison between specific humidity (lefthand axis, black) and tropospheric airmass-weighted air temperature (righthand axis, blue). Meteorology plays an important role for IAV in OH primary production, with OH primary production being strongly correlated with changes in specific humidity (q) (r = 0.90) which itself is strongly correlated with temperature (r = 0.96).

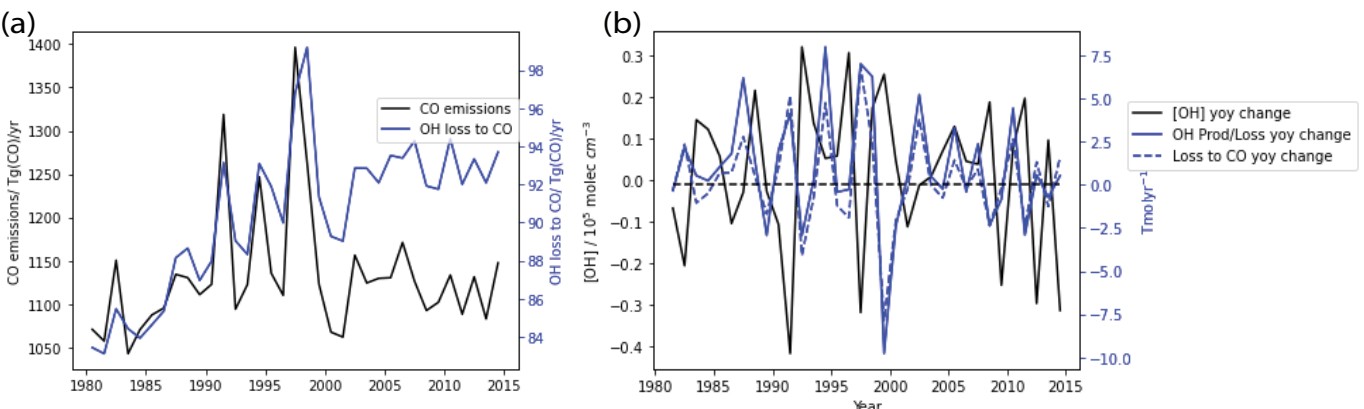

**Figure 10.** Plots of (a) OH loss to CO with CO emissions, and (b) the year-on-year changes in [OH] together with the year-on-year changes in net OH production and that of OH loss to CO. As seen in (a), CO emissions are positively correlated with OH loss to CO in the 'Base' run ($r = 0.69$). This thereafter drives overall IAV in total loss and production. As seen in (b), we see that OH year-on-year production/loss change (solid blue) is highly correlated with that of the OH loss to CO (dotted blue) with $r = 0.94$, suggesting that the OH loss to CO is indeed driving the IAV seen in total production/loss. The year-on-year change in production/loss, in turn, is anti-correlated with the year-on-year change in OH, with $r = -0.36$.

the OH loss to CO on the righthand axis (blue), we first notice that the OH year-on-year production/loss change (solid blue) is highly correlated with that of the OH loss to CO (dotted blue) with $r = 0.94$, suggesting that the OH loss to CO is indeed driving the IAV seen in total production/loss. The year-on-year change in production/loss, in turn, is anti-correlated with the

year-on-year change in OH, with $r = -0.36$.





## 4 Evaluation of Modelled CO and NO$_x$

### 4.1 CO: Comparison with Satellite Column and In Situ Surface Measurements

Fig. 11(a) shows the MOPITT and modelled CO column trends from 2001 to 2014, and Fig. 11(b) shows the input CO emission trends over the same time period. The MOPITT CO column generally sees significant negative trends throughout the spatial

domain, consistent with results from Yin et al. (2015). However, the modelled CO columns show significant trends positive trends above China and Iand with weaker positive trends in parts of Africa and the Middle-East and negative trends over most other parts of the world. The mixed modelled CO column trends mirror the CO emission trends, especially the increasing trend over China and South Asia, and are in poor agreement with trends derived from MOPITT. The MOPITT and modelled CO column trends thus show a poor agreement. Comparing Fig. 11(a) and (b), we see that the modelled CO column trends

exhibit a high spatial correlation with the input CO emission trends, and so the mismatch between observed and modelled CO column trends could point towards some deficiencies in the input emissions, which drive the high bias in CO column trends over China and South Asia and in turn leads to the general high bias globally due to transport from these regions especially via the prevailing Westerlies (Zheng et al., 2018). Zheng et al. (2019) suggest that emissions from the version of CEDS used in our study are inconsistent for China and South Asia. Whereas the version of CEDS used in this study (and other CMIP6

runs) suggests rapidly increasing anthropogenic CO emissions from China and South Asia, Zheng et al. (2019) found instead a decreasing trend for China and a modestly-increasing one for South Asia. Elguindi et al. (2020) compared regional bottom-up inventories, global bottom-up inventories which include CEDS, and top-down estimates, and they also found that CEDS (and other global bottom-up inventories) showed an increasing trend that was larger than seen in regional inventories and top-down emissions. They further point out that, for these countries which are experiencing rapid changes in their economies, technology

and environmental policies, the reason for biases in the global bottom-up inventories is that they may lack the latest data about regional activity and emission factor changes, and so in these cases using regional or top-down inventories might reduce biases.

There are various implications of the model-observations mismatch. Given what we understand about how CO affects modelled [OH] in the GFDL AM4.1 from our earlier analyses, we could suspect that, in places where we underestimate the decreasing trend, we could be underestimating the [OH] increase due to decreasing CO. Meanwhile, in those areas where we

model a CO increase when observations suggest a decrease instead, we may be modelling an [OH] decrease due to increasing CO in those regions, as opposed to an [OH] increase due to decreasing CO. Overall, this means that, globally, we could be underestimating the [OH] increase over the 2001-2014 period.

### 4.2 NO$_x$: Comparison of Tropospheric NO$_2$ Column with OMI

Fig. 12(a) shows the annual mean OMI tropospheric NO$_2$ column and modelled NO$_2$ column trends from 2005 to 2014, and

Fig. 12(b) shows the input NO$_x$ emission trends over the same time period. We see that there is general agreement between the emissions and the model tropospheric NO$_2$ column. Also, both show significant positive trends over South Asia and Eastern China, and significant negative trends over Eastern USA and Western Europe. These, in turn, are also consistent with the OMI tropospheric NO$_2$ column observations (the OMI observations over the remote ocean are likely to have large errors due to the



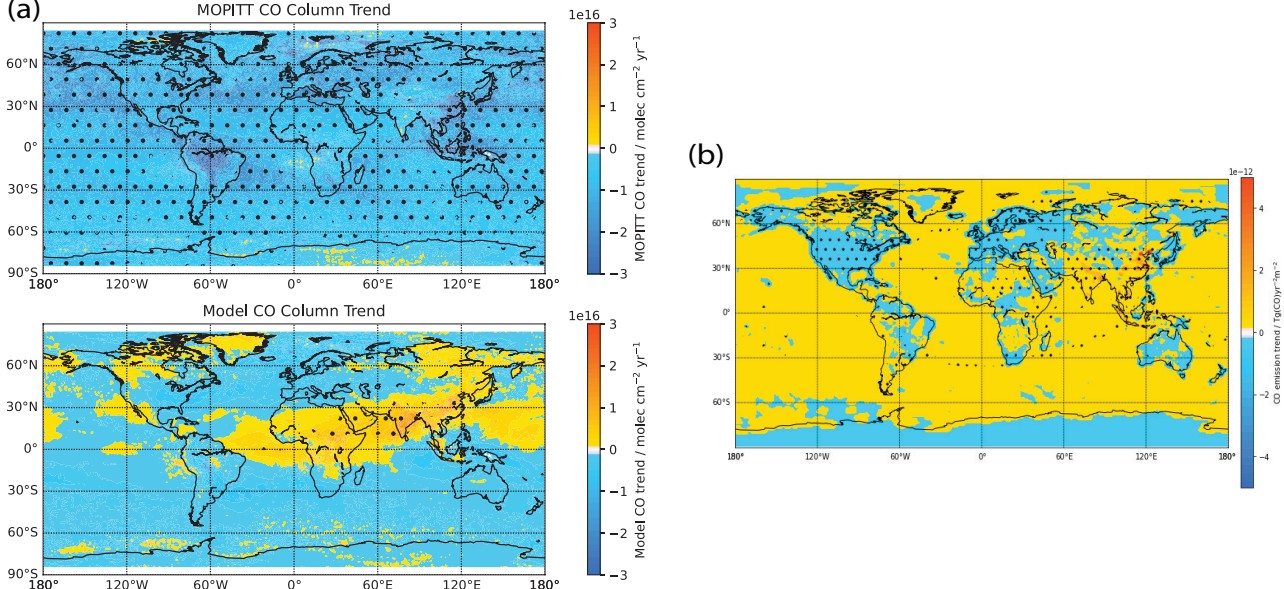

**Figure 11.** (a) Model comparison of annual mean CO column trends with MOPITT observations over the 2001-2014 period, and (b) CO emission trends over the 2001-2014 period. Trends and stippling are as per Fig. 3. The MOPITT CO column generally sees significant negative trends throughout the spatial domain. However, the modelled CO columns show significant trends positive trends above China and Iand with weaker positive trends in parts of Africa and the Middle-East and negative trends over most other parts of the world. The mixed modelled CO column trends mirror the CO emission trends, especially the increasing trend over China and South Asia, and are in poor agreement with trends derived from MOPITT. Comparing (a) and (b), we see that the modelled CO column trends exhibit a high spatial correlation with the input CO emission trends, and so the mismatch between observed and modelled CO column trends could point towards some deficiencies in the input emissions, which drive the high bias in CO column trends over China and South Asia and in turn leads to the general high bias globally due to transport from these regions especially via the prevailing Westerlies (Zheng et al., 2018).

observational detection limit.) From the literature, Miyazaki et al. (2017), who looked at an assimilation of multiple satellite datasets, including the OMI $NO_2$ column, obtained a global non-lightning $NO_x$ emission from 2005 to 2014 of a roughly constant value of $47.9 \mathrm{TgN/yr}$. This also agrees well with the emissions inventory used in our model, as shown in Fig. 1(c), where, even though we are slightly high-biased (about 8% higher), our global $NO_x$ emissions are also stable from 2005 to 2014. Overall, these findings lend confidence in our model $NO_x$ trends from 2005 to 2014.





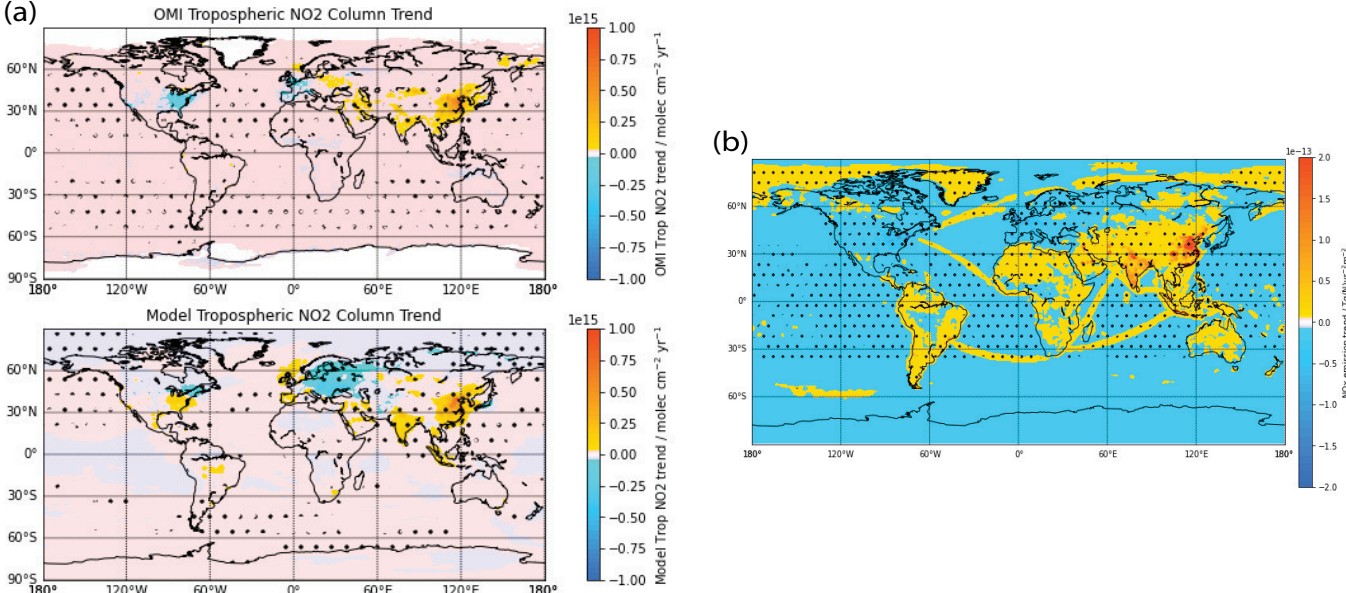

**Figure 12.** (a) Model comparison of annual mean tropospheric $NO_2$ column trends with OMI tropospheric $NO_2$ observations over the 2005-2014 period, and (b) $NO_x$ emission trends over the 2005-2014 period. Trends and stippling are as per Fig. 3. We see that there is general agreement between the emissions and the model tropospheric $NO_2$ column. Also, both show significant positive trends over South Asia and Eastern China, and significant negative trends over Eastern USA and Western Europe. These, in turn, are also consistent with the OMI tropospheric $NO_2$ column observations. Overall, these findings lend confidence in our model $NO_x$ trends from 2005 to 2014.

## 5   Conclusions

In this study, we systematically analyzed the sensitivity of [OH] to changes in drivers of OH over the 1980-2014 period using the GFDL AM4.1 model. We attribute the [OH] changes to changes in emissions and meteorology individually as opposed to a lumped approach adopted in the multimodel study by Stevenson et al. (2020). Such a decomposition allows for a clearer mechanistic understanding of the main driving factors of either the trend or IAV. In addition, we analysed the OH budget terms, similar to Zhao et al. (2020), tracing the individual emissions to changes in the various budget terms, which in turn affects

[OH].

We found that annual mean global tropospheric airmass-weighted [OH] has increased by $\sim 10\%$ compared to the 1998-2007 mean from 1980 to 2014, in agreement with multimodel comparisons of ESMs by Stevenson et al. (2020), and furthermore has increased by $\sim 5\%$ in 2014 compared to 1980, in agreement with multi-model studies such as from ACCMIP (Naik et al., 2013) and CCMI (Zhao et al., 2019). This modelled increasing [OH] trend, especially post-2007, is in contrast with the absence

of change (e.g., Nicely et al., 2020; Patra et al., 2021) or decreasing [OH] (e.g., Rigby et al., 2017; Turner et al., 2017) derived from observationally-constrained inversion methods.





In our model, the increasing trend in [OH] is caused by the net effects of increasing $NO_x$ emissions, which increases [OH] via both primary and secondary [OH] production, balanced by the increase in $CH_4$ concentrations which tend to consume OH. The combined effects of $NO_x$ emissions and $CH_4$ concentrations can account for the spatial distribution of the [OH] trends as well. These findings agree with other studies, such as Naik et al. (2013), who also suggested the importance of $NO_x$ and $CH_4$ in driving the modelled [OH] trend. Locally, CO emissions, meteorology and NMVOC emissions also play an important role in driving the increasing [OH] trend, but their effects average out on the global level. Meanwhile, the observed [OH] IAV is dominated by impacts from the IAV in biomass burning CO emissions as well as meteorology.

We also found that our model does a poor job of matching MOPITT total column CO trends over the 2001-2014 period. Given that modelled column CO trends were driven by input CO emission trends, this could in turn point towards some deficiencies in the input emissions. Zheng et al. (2019) further suggest that emissions from CEDS, which is the anthropogenic CO emissions dataset used in our study, are inconsistent for China and South Asia. Whereas CEDS suggests rapidly increasing CO from China and South Asia, Zheng et al. (2019) found instead a decreasing trend for China and a modestly-increasing one for South Asia. The increasing CO trends from China and South Asia, in turn, leads to higher CO levels. Additionally, we found that the modelled tropospheric $NO_2$ column trends qualitatively agrees with OMI satellite tropospheric $NO_2$ column trends over the 2005-2014 period. Thus, overall, the underestimated declining trend in CO emissions in our model could mean that the actual modelled [OH] increase is larger than what was currently modelled.

## 5.1 Implications

Overall, based on the current set-up, the AM4.1 models an increase in tropospheric [OH] from 1980 to 2014. This is even in the backdrop of the increase in $CH_4$ throughout the period. As seen in Fig. 13, this causes the $CH_4$ lifetime with respect to OH to decrease in the 'Base' run by about 10%. In the absence of other changes, one would expect the increase in $CH_4$ to reduce OH, thereby further prolonging the $CH_4$ lifetime. Instead, we see that the increase in $NO_x$ together with the stalling of CO has led to a greater increase in OH than would be expected by the increase in $CH_4$, such that $CH_4$ lifetime still continues to decrease. This helps to slow the accumulation of $CH_4$ in the atmosphere. However, despite the lower lifetime, $CH_4$ concentrations have still continued to steadily increase. As a result of our modelled increasing [OH], this suggests that studies trying to derive $CH_4$ emissions from observed $CH_4$ concentrations will underestimate $CH_4$ emissions if they do not take into account the increasing [OH].

Also, in the future, should aggressive air quality policies cause a reduction in $NO_x$ emissions, this could cause the [OH] to decrease, thereby further accelerating the buildup of $CH_4$ in the atmosphere. On the other hand, if CO emissions also decrease concomitantly, this could offset the $NO_x$ reduction effects on [OH]. Future work could involve looking at how OH evolves under future scenarios, such as the Shared Socioeconomic Pathways (SSPs) formulated as part of CMIP6. In particular, future work could focus on whether $NO_x$ and CO will still play a dominant role in the future, under different scenarios of climate change as well as emissions reductions. The recent COVID-19 related large reduction in emissions in cities across the world have provided a glimpse of what could happen in future scenarios. For example, Laughner et al. (2021) found that the decrease in $NO_x$ emissions in 2020 led to a decrease in ozone which thereby led to a 2-4% decrease in global [OH], and this could have





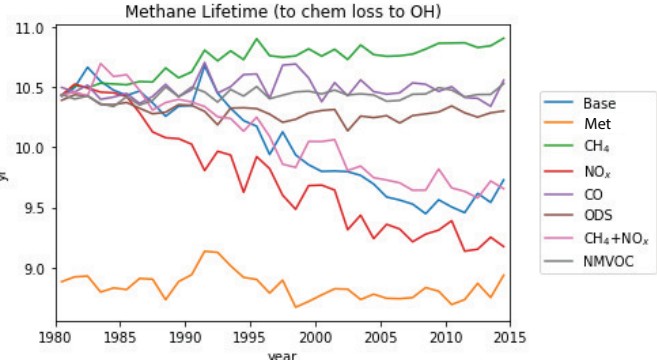

**Figure 13.** $CH_4$ lifetime with respect to oxidation by OH for the different model runs. $CH_4$ lifetime has decreased in the 'Base' run even as $CH_4$ concentrations have increased, driven by [OH] increases due to increasing $NO_x$ emissions.

contributed to the large [$CH_4$] growth rate that year. This was further corroborated by Stevenson et al. (2022) and Peng et al. (2022) who both found that about half of the large [$CH_4$] growth rate was attributed to the decline in [OH] due to declining $NO_x$. Peng et al. (2022) also additionally showed that the effect of declining $NO_x$ emissions which led to decreased [OH] overwhelmed the impacts of the decline of other SCLFs like CO emissions. $NO_x$ emissions have since largely returned back
to pre-pandemic levels, and this could drive [OH] increases again. In the backdrop of the anthropogenic emission changes, the pandemic years have also seen many large wildfire events as well, and the associated biomass burning emissions could also impact [OH]. Overall, these could be interesting test cases to explore in ESMs with interactive chemistry.

In our paper, we explored the role of meteorology and input emission inventories in driving the [OH] trend during the 1980-2014 period, but, as Murray et al. (2021) points out, there are many other factors within models that could be important
in driving inter- and intra-model [OH] variations, with key factors being the details of the implemented chemical scheme, which has implications on oxidation of VOCs into CO and $NO_x$ lifetime, as well as other physical parameterizations, such as lightning $NO_x$ altitude which also affects $NO_x$ lifetime. The importance of understanding the CO budget drivers and potential biases is further underscored by the existing biases present in the GFDL AM4.1, such as in the seasonal mean CO column Horowitz et al. (2020) and the CO column trend. Nonetheless, given that we have identified various input emission drivers as
playing a key role in driving the increasing [OH] trend over the 1980-2014 period, and other models participating in CMIP6 also have likely used the same anthropogenic emission inventories, our study could also serve as a motivation to do a similar sensitivity analysis in other CCMs, such as the other ESMs studied in Stevenson et al. (2020). This could help elucidate the role of emissions in driving the multi-model mean trend, and potentially further emphasize the importance of accurate short-lived climate forcer emission inventories for both climate and air quality projections (Smith et al., 2022).

*Code and data availability.* The code and data used in this paper are available upon request.



*Author contributions.* GC wrote the text and performed the main analysis. VN and LWH helped to conceptualize the study and provided comments.

*Competing interests.* The contact author has declared that none of the authors has any competing interests.

*Acknowledgements.* We acknowledge GFDL HPC resources without which the AM4.1 simulations would not have been possible. We also
acknowledge Jian He who provided additional comments as well as help with the MOPITT and OMI data used in this study. We also
acknowledge the two internal reviewers Chloe Gao and Meiyun Lin for their useful comments.





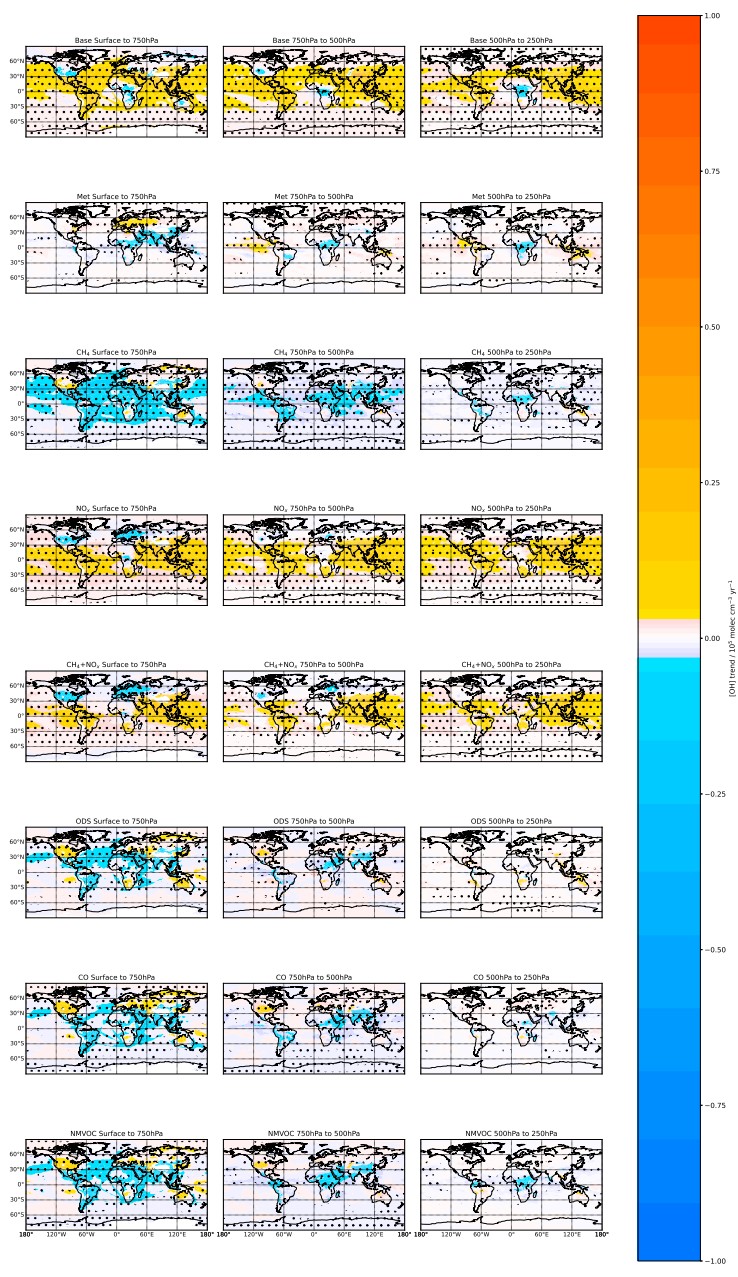

**Figure A1.** Spatial airmass-weighted tropospheric [OH] trends at different pressure levels for the different runs. Trends and stipples are as per Fig 3.



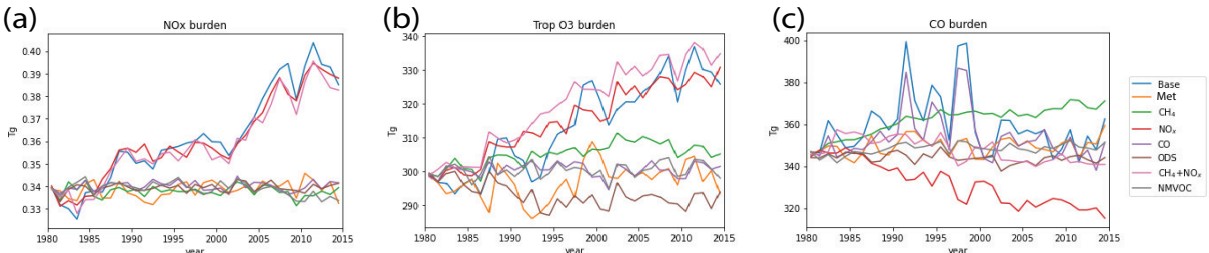

**Figure A2.** (a) Tropospheric O$_3$, (b) NO$_x$ and (c) CO burdens for the different model runs.

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
