# Peer review of "Exploring the Drivers of Tropospheric Hydroxyl Radical Trends in the GFDL AM4.1 Atmospheric Chemistry-Climate Model"

_Atmospheric Chemistry and Physics, 2023_

## Author Comment (AC1)

This manuscript presents an examination of hydroxyl radical trends, variability, and sensitivity from the GFDL model AM4.1 for 1980-2014. In addition to a "Base" run and a "Met" run in which all emissions are fixed to 1980 levels, sensitivity simulations are also performed in which emissions for individual species (NOx, CH4, CO, NMVOCs, and ODSs) are fixed to 1980 to isolate spatial and temporal effects on OH abundance. Results suggest that global mean OH concentration has increased by ~5%, mainly due to the competing effects of increasing NOx and CH4. Model validation against OMI NO2 and MOPITT CO is performed, revealing that NO2 compares reasonably well while modeled CO trends compare poorly against observations (which reflects more on the emissions inventory than on the model). Overall, I consider this to be a nice analysis that makes a solid contribution to the literature surrounding OH concentrations at the global scale. Sensitivity simulations like the ones performed here are valuable for gleaning information about the drivers of OH variability, with interesting, if perhaps somewhat expected, conclusions found in the spatial and temporal details of the various analyses of the simulations. I consider the comparison to observations to be sufficient for this study -— there are always additional datasets that can be compared against, but for the species examined and the motivation of this work, the two included make sense. Prior literature is well cited, and the present work is well contextualized with comparisons to the results of other studies. The article is well within the scope of ACP, and, after addressing a number of comments included below, I would consider it a good candidate for publication.

**We thank the reviewer for their overall positive feedback as well as specific comments. We address each of the specific comments below:**

Specific comments:
Table 1: Curious that a wavelength cutoff of 310 nm is used for O3->O1D photolysis; most other models (e.g., Lelieveld et al., 2016, which you compare to throughout this manuscript) use 330 nm due to the small contribution from the quantum yield tail – see, e.g., Armerdling et al., 1995: https://pubs.acs.org/doi/pdf/10.1021/j100010a025. Any idea, or ability to quantify, how much this might affect total primary production in your results?

**We have changed it to say '<330 nm'. In our model, we use the Fast-JX version 7.1 scheme (Wild et al., 2000; Bian and Prather, 2002) to calculate photolysis rates which includes this larger wavelength tail, and does not have a fixed cutoff for wavelength.**

L225 / Fig. 2b: While the increase in [OH] in the lower troposphere is largest in absolute terms, [OH] values drop in the UT just as a result of pressure. Would be informative to also see this plot in units of pptv.

The altitudinal profile of [OH] is reported in airmass-weighted concentrations of molecules per volume following other papers reporting [OH] such as Zhao et al. (2019). However, we have now also plotted the percentage change of 2005-2014 mean compared to 1980-1989 mean at each altitude in Fig. 2(b), instead of just presenting the absolute values (this figure is the same whether we use volume mixing ratios or airmass-weighted concentrations of molecules per volume).

[Figure]

L235: I'm curious if the authors see any issue with treating CH4 as a surface boundary condition and making conclusions like "CH4 caused a negative trend in [OH]". Especially since it's a problem of "the chicken and the egg" and feedbacks between OH and CH4 are notably missing at the surface, isn't causation particularly difficult to attribute in this case? Since models are generally not set up to do CH4 fluxes, the model configuration here is understandable; perhaps just worth a note of caution in the text.

We acknowledge the comment and are also aware of the drawbacks of the current modelling setup. However, it is clear from our simulations that the imposed changing methane concentrations are causing significant OH trends. While the feedbacks are missing on the surface, methane and OH are still allowed to interact throughout the rest of the troposphere. Forcing the model with surface methane emissions instead of surface concentrations will likely further amplify these significant OH trends due to the methane self-feedback. In our paper, we have taken more caution to emphasize that we are not using methane fluxes. We have added a paragraph in the 'Implications' section to say 'We acknowledge that CH4 concentrations are prescribed on the surface in the current model set-up, so this can lead to an underestimation of the surface chemical feedbacks. Including the surface feedbacks would likely amplify the modelled effects of CH4 on [OH]. This should be further investigated in an emissions-driven run.'

Figures 11 and 12: For both panels (b), does this indicate a non-zero emissions trend over the oceans? I don't see why a trend in emissions for either CO or NO2 should occur, besides for shipping lanes perhaps, but I would expect from the color bar that a zero trend should be depicted as white.

**The non-zero emissions trend over the oceans for CO comes from shipping. The non-zero-emissions trend over the oceans for NOx comes not only from shipping but also from aircraft as well.**

Technical corrections:
L34: "tropospheric" misspelled
**Done.**
L86: "increasing" should be "increase"
**Done.**
L167: Check punctuation; period should be comma.
**Done.**
L203: should be "as well as"
**Done.**
L227: sensitivity misspelled
**Done.**
L243: "the" or "this" should be removed
**Done.**
Figure 2: in panel d), I think the purple bar lost part of its label (should be CH4+NOx? I only see "+NOx")
**Done.**
L296: should be "out of"
**Done.**
Figure 6: Text on panels c and d should be increased in size
**Done.**
L340: "increases" misspelled
**Done.**
L362: "flux" here is a bit hard to decipher, please clarify
**Done.**
L386: "trends" at beginning of line should be removed; is "Iand" a typo
**Done.**
Figure 11 caption: same as above. Also, I'd suggest avoiding repetition between figure captions and text.
**Thank you for your suggestion. We have revised it.**
L445: "agrees" should be "agree"
**Done.**
L469: Should this be "SLCF" instead? Defined?
**Thank you for pointing this out. We have revised it.**
L479: should be "(Horowitz et al., 2020)" all in parentheses?
**Thank you for pointing this out. We have revised it.**

**Thank you for the technical corrections, they have all been addressed and/or amended.**

**References**

Bian, H. and Prather, M. J.: Fast-J2: Accurate Simulation of Stratospheric Photolysis in Global Chemical Models, Journal of Atmospheric Chemistry, 41, 281–296, https://doi.org/10.1023/a:1014980619462, 2002.

Wild, O., Zhu, X., and Prather, M. J.: Fast-J: Accurate Simulation of In- and Below-Cloud Photolysis in Tropospheric Chemical Models, Journal of Atmospheric Chemistry, 37, 245–282, https://doi.org/10.1023/a:1006415919030, 2000.

Zhao, Y., Saunois, M., Bousquet, P., Lin, X., Berchet, A., Hegglin, M. I., Canadell, J. G., Jackson, R. B., Hauglustaine, D. A., Szopa, S., Stavert, A. R., Abraham, N. L., Archibald, A. T., Bekki, S., Deushi, M., J ckel, P., Josse, B., Kinnison, D., Kirner, O., Mar cal, V., O'Connor, F. M., Plummer, D. A., Revell, L. E., Rozanov, E., Stenke, A., Strode, S., Tilmes, S., Dlugokencky, E. J., and Zheng, B.: Intermodel comparison of global hydroxyl radical (OH) distributions and their impact on atmospheric methane over the 2000–2016 period, Atmospheric Chemistry and Physics, 19, 13 701–13 723, https://doi.org/10.5194/acp-19-13701-2019, 2019.

---

## Author Comment (AC2)

Chua et al use the GFDL CCM to understand the modeled trend in tropospheric OH over 1980 -2014 and, through using various sensitivity runs, to tease out the relative importance of the different OH drivers on these trends. Ultimately, they find that the increasing trend in NOX emissions over the period, along with the increase in CH4 abundance, have the largest impact on global OH trends. CO and meteorology, through it's impact on water vapor abundance, primarily impact interannual variability, although they can affect trends regionally. They also compare modeled trends in CO and NO2 to observed trends from MOPITT and OMI, respectively, finding that the model replicates NO2 trends well but fails to capture CO accurately. They attribute errors in the CO trends to errors in the emissions. Overall, this is a well-written paper that incrementally advances our understanding of OH variability. It is suitable for publication in ACP once the minor revisions below are addressed.

**We thank the reviewer for their overall positive feedback as well as specific comments. We address each of the specific comments below:**

Line 125: Since lightning NOX is so important for OH production, a few more details about how lightning NOX is calculated in the model should be included.
**We have included more details of how lightning NOx is calculated in the model in the updated manuscript. We now have included the following text: 'As described in Horowitz et al. (2020), lightning NOx emissions are calculated interactively as a function of subgrid convection, as diagnosed by the double plume convection scheme described by Zhao et al. (2018b). The lightning NOx source is calculated as a function of convective cloud-top height, following the parameterization of Price et al. (1997), and is injected with the vertical distribution of Pickering et al. (1998).'**

Line 127: Are the surface concentrations of CH4 and the other species set by latitude? What dataset do you use to constrain the values?
**We use Meinshausen et al. (2017), and use a global mean value for the surface as opposed to setting by latitude.**

Line 130: Should be "A summary of historical emissions … is shown in Fig 1."
**Thank you for the suggestion. We have made the change.**

Line 166: You say that you don't need to evaluate the CH4 since surface values are set as a boundary condition, but this does not necessarily translate to CH4 being correct aloft or even at the surface, since I'm assuming you're using latitude bands to set the surface concentration. Since CH4 plays such an important role in your results, seeming to be second only in importance to anthropogenic NOX on a global scale, some discussion of how errors in CH4 could affect your results is warranted.

**We have added a paragraph in the 'Implications' section to say 'We acknowledge that CH4 concentrations are prescribed on the surface in the current model set-up, so this can lead to an underestimation of the surface chemical feedbacks. Including the surface feedbacks would likely amplify the modelled effects of CH4 on [OH]. This should be further investigated in an emissions-driven run.'**

Line 198: Why aren't you using the most recent OMI NO2 retrieval (v4.0) (Lamsal et al, 2021)? Changes in the air mass factors for the new retrieval have led to some large changes in the retrievals, particularly over highly polluted regions (see Fig. 10 in Lamsal, for example). Are these changes irrelevant for the trends you are studying?
**We have revised the figure to use OMI NO2 retrieval (v4.0) that has been processed by Goldberg et al. (2021).**

[Figure]

Line 220: Is He et al (2020) using the same simulation you discuss here, or one similar enough in configuration that the OH trends can be compared? Also, in the citations, you list the version of He et al (2020) from ACPD. That should be updated to the finalized version.
**He et al. (2020) use surface methane emissions that have been optimized so as to reproduce the observed surface methane concentrations. Their OH trends are comparable to ours (for example, in their figure 6). We have also updated the citation.**

Line 300: The dip in 1992 is also evident in the met run, indicating that, for this case, CO isn't necessarily the main/driving factor. Assuming your simulation includes the effects of the Pinatubo eruption on the stratosphere, isn't this a more likely explanation for that particular dip, at least in part? There's no need to get into a discussion about this but maybe just removing the reference to 1992 would simplify things.

**We thank the reviewer for the suggestion, and have implemented it.**

Figures 4 and 6: For all panels in Figure 4 and for panels c and d of Figure 6, most of the text is illegible. Please increase the font size.

**We thank the reviewer for the suggestion, and have implemented it.**

Line 340: Should be "increases" not "increses".

**We have made the suggested changes.**

Line 385 – 386: Should say "significant positive trends". Also, I think Iand is supposed to be India?

**We thank the reviewer for pointing out the mistake and we have made the suggested changes.**

Section 4.1: Since your model results suggest that CO affects global OH more through IAV than through trends, I think it also warrants some discussion on how well the model captures the CO IAV as compared to MOPITT. Otherwise, I think the MOPITT evaluation section is sufficient in highlighting the potential limitations of the impact of the modeled CO on this analysis.

**We have included a comparison of model and MOPITT area-weighted rolling 12 month mean CO column from 60S to 60N. We found that the Pearson correlation is high (r=0.62, and r=0.76 from the detrended series) which suggests that the SST-driven run captures the IAV well. Also, this supports the earlier findings in the paper that the model underestimates the CO column trends compared to MOPITT. The trend (calculated using the Theil-Sen method) for MOPITT observations is -0.010 x $10^{18}$ molecules cm$^{-2}$ yr$^{-1}$ , which is about 5 times that for MOPITT which is -0.021 x $10^{18}$ molecules cm$^{-2}$ yr$^{-1}$. We have updated Fig. 11 with these additional figures and have added a discussion on the IAV comparison.**

[Figure]

Line 444: Should be "The increasing CO trends … lead to higher CO levels."

**We have made the change.**

Figure 13: Something seems off about the methane lifetime for the "Met" run. If I'm understanding correctly, for that simulation, all anthropogenic emissions were held to 1980 values, so while it's understandable that there would be large differences by the end of the simulation, it seems unrealistic that, in 1981, the CH4 lifetime would differ by more than 1.5 years from the baseline simulation.

**We thank the reviewer for pointing the error out; there was an error in the calculation on our end, and it has been corrected. This is the updated Fig. 13.**

[Figure]

**References**

Goldberg, D. L., Anenberg, S. C., Lu, Z., Streets, D. G., Lamsal, L. N., McDuffie, E. E., and Smith, S. J.: Urban NOx emissions around the world declined faster than anticipated between 2005 and 2019, Environmental Research Letters, 16, 115 004, https://doi.org/10.1088/1748-9326/ac2c34, 2021.

He, J., Naik, V., Horowitz, L. W., Dlugokencky, E., and Thoning, K.: Investigation of the global methane budget over 1980–2017 using GFDL-AM4.1, Atmospheric Chemistry and Physics, 20, 805–827, https://doi.org/10.5194/acp-20-805-2020, 2020.

Horowitz, L.W., Naik, V., Paulot, F., Ginoux, P. A., Dunne, J. P., Mao, J., Schnell, J., Chen, X., He, J., John, J. G., Lin, M., Lin, P., Malyshev, S., Paynter, D., Shevliakova, E., and Zhao, M.: The GFDL Global Atmospheric Chemistry-Climate Model AM4.1: Model Description and Simulation Characteristics, Journal of Advances in Modeling Earth Systems, 12, https://doi.org/10.1029/2019ms002032, 2020.

Pickering, K. E., Wang, Y., Tao, W.-K., Price, C., and Müller, J.-F.: Vertical distributions of lightning NOx for use in regional and global chemical transport models, Journal of Geophysical Research: Atmospheres, 103, 31 203–31 216, https://doi.org/10.1029/98jd02651, 1998.

Price, C., Penner, J., and Prather, M.: NOx from lightning: 1. Global distribution based on lightning physics, Journal of Geophysical Research: Atmospheres, 102, 5929–5941, https://doi.org/10.1029/96jd03504, 1997.

Zhao, M., Golaz, J.-C., Held, I. M., Guo, H., Balaji, V., Benson, R., Chen, J.-H., Chen, X., Donner, L. J., Dunne, J. P., Dunne, K., Durachta, J., Fan, S.-M., Freidenreich, S. M., Garner, S. T., Ginoux, P., Harris, L. M., Horowitz, L.W., Krasting, J. P., Langenhorst, A. R., Liang, Z., Lin, P., Lin, S.-J., Malyshev, S. L., Mason, E., Milly, P. C. D., Ming, Y., Naik, V., Paulot, F., Paynter, D., Phillipps, P.,Radhakrishnan, A., Ramaswamy, V., Robinson, T., Schwarzkopf, D., Seman, C. J., Shevliakova, E., Shen, Z., Shin, H., Silvers, L. G., Wilson, J. R., Winton, M., Wittenberg, A. T., Wyman, B., and Xiang, B.: The GFDL Global Atmosphere and Land Model AM4.0/LM4.0: 1. Simulation Char acteristics With Prescribed SSTs, Journal of Advances in Modeling Earth Systems, 10, 691–734, https://doi.org/10.1002/2017ms001208, 2018.